# Financial incentives and deposit contracts to promote HIV retesting in Uganda: A randomized trial

Gabriel Chamie[1]*, Dalsone Kwarisiima[2], Alex Ndyabakira[2], Kara Marson[1], Carol S. Camlin[1], Diane V. Havlir[1], Moses R. Kamya[2,3], Harsha Thirumurthy[4,5]

**1** University of California, San Francisco, San Francisco, California, United States of America, **2** Infectious Diseases Research Collaboration, Kampala, Uganda, **3** Makerere University, Kampala, Uganda, **4** Center for Health Incentives and Behavioral Economics, University of Pennsylvania, Philadelphia, Pennsylvania, United States of America, **5** Perelman School of Medicine, University of Pennsylvania, Philadelphia, Pennsylvania, United States of America

* Gabriel.Chamie@ucsf.edu

**Data Availability Statement:** Data is held in a public repository and is available and publicly accessible at the following url: https://dataverse.

## Abstract

### Background

Frequent retesting for HIV among persons at increased risk of HIV infection is critical to early HIV diagnosis of persons and delivery of combination HIV prevention services. There are few evidence-based interventions for promoting frequent retesting for HIV. We sought to determine the effectiveness of financial incentives and deposit contracts in promoting quarterly HIV retesting among adults at increased risk of HIV.

### Methods and findings

In peri-urban Ugandan communities from October to December 2018, we randomized HIV−negative adults with self-reported risk to 1 of 3 strategies to promote HIV retesting: (1) no incentive; (2) cash incentives (US\$7) for retesting at 3 and 6 months (total US\$14); or (3) deposit contracts: participants could voluntarily deposit US\$6 at baseline and at 3 months that would be returned with interest (total US\$7) upon retesting at 3 and 6 months (total US\$14) or lost if participants failed to retest. The primary outcome was retesting for HIV at both 3 and 6 months. Of 1,482 persons screened for study eligibility following community-based recruitment, 524 participants were randomized to either no incentive (*N* = 180), incentives (*N* = 172), or deposit contracts (*N* = 172): median age was 25 years (IQR: 22 to 30), 44% were women, and median weekly income was US\$13.60 (IQR: US\$8.16 to US\$21.76). Among participants randomized to deposit contracts, 24/172 (14%) made a baseline deposit, and 2/172 (1%) made a 3-month deposit. In intent-to-treat analyses, HIV retesting at both 3 and 6 months was significantly higher in the incentive arm (89/172 [52%]) than either the control arm (33/180 [18%], odds ratio (OR) 4.8, 95% CI: 3.0 to 7.7, *p* < 0.001) or the deposit contract arm (28/172 [16%], OR 5.5, 95% CI: 3.3 to 9.1, *p* < 0.001). Among those in the deposit contract arm who made a baseline deposit, 20/24 (83%) retested at 3 months; 11/24 (46%) retested at both 3 and 6 months. Among 282 participants who retested

harvard.edu/dataset.xhtml?persistentId=doi:10.
7910/DVN/PXCL0W.

**Funding:** This work was funded through a grant to
GC and HT (Grant number: R01MH105254) from
the National Institute of Mental Health (NIMH, url:
https://www.nimh.nih.gov) at the National
Institutes of Health. The funders had no role in
study design, data collection and analysis, decision
to publish, or preparation of the manuscript.

**Competing interests:** The authors have declared
that no competing interests exist.

**Abbreviations:** ART, antiretroviral therapy; GDP,
gross domestic product; HIVST, HIV self-testing;
IQR, interquartile range; MoH, Ministry of Health;
OR, odds ratio; PrEP, preexposure prophylaxis; SD,
standard deviation.

for HIV during the trial, three (1%; 95%CI: 0.2 to 3%) seroconverted: one in the incentive
group and two in the control group. Study limitations include measurement of retesting at
the clinic where baseline enrollment occurred, only offering clinic-based (rather than com-
munity-based) HIV retesting and lack of measurement of retesting after completion of the
trial to evaluate sustained retesting behavior.

## Conclusions

Offering financial incentives to high-risk adults in Uganda resulted in significantly higher HIV
retesting. Deposit contracts had low uptake and overall did not increase retesting. As part of
efforts to increase early diagnosis of HIV among high-risk populations, strategic use of
incentives to promote retesting should receive greater consideration by HIV programs.

## Trial registration

clinicaltrials.gov: NCT02890459.

---

## Author summary

### Why was this study done?

- Frequent retesting for HIV among persons who face a high risk of HIV acquisition is
  essential for protecting health and preventing HIV transmission.

- In high HIV prevalence settings, there is a need for interventions that can increase HIV
  retesting rates among people who have a higher risk of HIV acquisition.

- Financial incentives and deposit contracts have been effective in promoting health
  behaviors, but there is no evidence on whether they can increase the likelihood of HIV
  retesting.

### What did the researchers do and find?

- We conducted a randomized controlled trial in Uganda to determine whether offering
  cash incentives or deposit contracts increases retesting for HIV at 3 and 6 months after
  a negative HIV test among 524 adults with risk factors for HIV.

- Participants in the deposit contract group were given an opportunity to make a cash
  deposit as a commitment to future retesting, with the deposit returned with interest
  upon retesting.

- The cash incentives group was significantly more likely than the control group to retest
  for HIV at 3 and 6 months (52% versus 18%).

- Fourteen percent of participants in the deposit contract group made a deposit, and
  although these participants were very likely to retest for HIV, overall retesting rates at 3
  and 6 months were similar in the deposit contract group and control group (16% and
  18%).

**What do the findings mean?**

- Cash incentives should be considered as a highly effective intervention for increasing routine retesting for HIV among adults at risk for HIV infection.

- Low uptake of deposit contracts is likely to limit the success of this approach in increasing retesting rates.

## Introduction

Frequent retesting for HIV among persons at increased risk of infection is critical to HIV control efforts. With routine retesting and early HIV diagnosis, there are greater opportunities for HIV treatment with antiretroviral therapy (ART) to reduce HIV-associated morbidity and eliminate onward HIV transmission [1,2]. Similarly, as novel forms of prevention emerge, retesting offers the opportunity for early introduction to a growing number of prevention modalities [3].

The World Health Organization recommends annual retesting among sexually active adults living in settings with generalized HIV epidemics, with more frequent retesting (every 3 to 6 months) for people based on individual risk factors [4]. In Uganda, the Ministry of Health (MoH) recommends HIV retesting every 3 months for key populations [5]. Yet published data suggest that HIV retesting rates remain suboptimal in sub-Saharan Africa [6,7]. Relatively few adults meet the annual testing recommendation and health programs face challenges in encouraging people to retest [8,9]. Furthermore, there are few evidence-based interventions designed specifically to promote frequent retesting for HIV [10]. Low retesting uptake may be due, in part, to perceptions that retesting is unnecessary if a person continues to feel healthy and recently tested HIV negative [11].

Like some other health behaviors, HIV retesting may also be hindered by biases in human decision-making such as present bias, a tendency to place disproportional weight on near-term rather than long-term costs and benefits [12]. Studies have also found that scarcity of income may amplify people's tendency to discount the future and worsen their ability to process health information [13,14], emphasizing the challenge programs face when promoting prevention behaviors with long-term benefits but few obvious short-term gains. Financial incentives, which have been effective in increasing one-time HIV testing and other health behaviors [15–17], offer one way to overcome present bias and the tendency to delay HIV testing. Since individuals may view retesting as a costly or inconvenient behavior of limited value, incentives may motivate individuals to seek regular HIV testing.

Behavioral economics research indicates that incentives can be more effective if they leverage loss aversion: people's tendency to place greater psychological emphasis on monetary losses than monetary gains of similar value [18]. Deposit contracts do exactly this by enabling individuals to voluntarily commit to a health goal by making a deposit that is retrieved only if they achieve the goal [19]. Deposit contracts have largely been implemented in middle- and high-income countries [20–22]. Studies have found that although relatively few people make deposits, their effectiveness may be high among those who make deposits [20,23]. There have been few evaluations of deposit contracts in low-income countries, where poverty may limit people's ability to make deposits. In a prior pilot study, however, we found that offering

deposit contracts to promote retesting for HIV among at-risk adults in Uganda was feasible and acceptable [24].

In this study, we evaluated the effectiveness of financial incentives and deposit contracts in promoting quarterly HIV retesting among HIV–negative persons at increased HIV risk in a peri-urban Ugandan community.

## Methods

We conducted a 3-group randomized controlled trial to determine the effectiveness of financial incentives and deposit contracts for quarterly retesting among HIV–negative adults at increased risk of HIV infection (NCT:02890459). The study was conducted in peri-urban towns in Ibanda District, southwestern Uganda, where adult HIV prevalence is 5.1% [25].

In September 2018, we held meetings with local health officials and community representatives to identify venues frequented by key populations, as defined by the Uganda MoH, including sex workers, transport workers, and people in serodifferent relationships [5]. As described elsewhere, the venues identified included bars associated with commercial sex work, businesses associated with transactional sex, and transportation hubs that included high-risk men [26]. Study staff visited these venues over 3 months and distributed 1,777 recruitment cards inviting adults in both English and Runyankole to come to a local government-run clinic the following day for a free health evaluation that included HIV testing, hypertension, diabetes, and malaria screening. The cards indicated that those who came for an evaluation would receive a one-time cash transfer of 10,000 Ugandan Shillings (USh) (US$2.70 in 2018) for reimbursement of travel expenses to reach the clinic.

Individuals who came for an evaluation were eligible for the study if they were aged 18 to 59 years, tested negative for HIV, and reported at least one of the following risk factors in the prior 12 months: (i) >1 partner; (ii) a known HIV–infected partner; (iii) a history of a sexually transmitted infection; or (iv) paid or received money or gifts in exchange for sex. We excluded participants who reported an intention to move away from the community for ≥4 of the 6 months following recruitment, were unwilling to retest for HIV in the future, or had tested for HIV ≥3 times in the past 12 months. Rapid HIV antibody testing was done using test kits and a serial testing algorithm based on Uganda MoH guidelines [5]. Similar HIV testing procedures were used when retesting participants during the study. Individuals who tested HIV–positive during recruitment were provided same-day linkage to care and ART. Distribution of recruitment cards continued until target enrollment was reached.

### Procedures

Eligible adults who provided written informed consent were administered a baseline questionnaire (including questions about demographics, socioeconomic status and health, and sexual behavior, including HIV risk and testing behaviors) and randomized (1:1:1, by block randomization, stratified by sex, with block size = 9 and allocation sequence computer-generated prior to trial initiation; participants then selected a randomization "scratch off" card from 9 cards presented by staff, with replacement of each card taken with another card from the next block) to 3 groups: financial incentives, deposit contracts, or control. Participants in the financial incentives group were told they would receive a payment of 25,000 USh (approximately US$7 in 2018) in cash if they returned to the clinic and retested for HIV at 3 months, and the same amount if they retested at 6 months. For context, in 2018, annual gross domestic product (GDP) per capita was US$770 (US$2.11 per day) in Uganda [27]. Incentives for 6-month retesting were not conditional on having retested at 3 months (Fig 1).

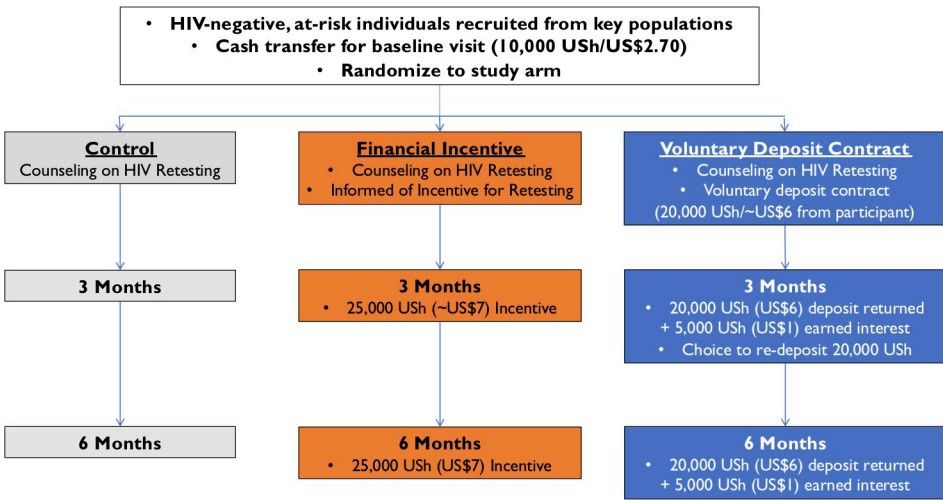

**Fig 1. Randomized controlled trial design and study interventions.**

Participants in the deposit contract group were given the option to voluntarily make a baseline deposit of 20,000 Ush (approximately US$6) to commit to retesting at 3 months. The deposit was not mandatory, and participants could retest whether or not they made a deposit. Participants in this group were told that if they returned for retesting in 3 months, they would be returned the 20,000 Ush deposit and earn an additional 5,000 Ush (approximately US$1) in interest. Those retesting at 3 months (regardless of having made a deposit) were given the option of reentering a deposit contract with the same terms for retesting at 6 months. Staff informed participants that they would lose their deposit if they did not come for retesting during the subsequent, prespecified retesting period. We chose the amount of money (approximately US$7) offered for both the financial incentive and the deposit plus interest based on recommendations from a community advisory board.

Participants in the control group did not receive any incentives to retest for HIV. In all groups, study staff counseled participants on the benefits of regularly retesting for HIV. Study staff informed participants they had 1-month windows for retesting (3 to 4 months and 6 to 7 months post-randomization) and provided a one-time phone call reminder for all participants who had not tested 3 weeks into the 1-month retesting windows at 3 and 6 months. Study staff administered a brief follow-up questionnaire to participants who retested at 3 and 6 months to inquire about HIV risk behaviors, perception of HIV risk, and reasons for retesting.

## Outcomes

The primary outcome was HIV retesting at the study-designated clinic at both 3 and 6 months. Secondary outcomes included HIV retesting at 3 months, retesting at 6 months, retesting among those who made deposits, and seroconverting to HIV antibody positive (see Supporting information, S1 Text: CONSORT checklist; and S2 Text: Study protocol).

## Statistical analyses

We estimated that with a sample size of 525 participants, there would be >80% power (alpha = 0.05, 2-sided) to detect a difference of ≥15% in retesting rates in each of the intervention groups compared to control. Descriptive statistics were used to present baseline characteristics, including means, standard deviations (SD), medians, and interquartile ranges (IQR).

We compared the proportion of participants in each group who achieved the primary and secondary outcomes using 2-tailed $\chi^2$ tests. We also performed logistic regression analyses to report unadjusted odds ratios (ORs) for retesting in the financial incentive arm versus control, deposit contracts versus control, and financial incentives versus deposit contracts. For retesting outcomes, we performed intent-to-treat analyses. In a secondary analysis, we used a standard instrumental variables approach (two-stage least squares) to estimate the causal effect of making a deposit on HIV retesting at 3 and 6 months [28]. Statistical analyses were performed using Stata version 15 (StataCorp).

## Ethical statement

All participants provided written informed consent in their preferred language (English or Runyankole). The Makerere University School of Medicine Research and Ethics Committee, the Uganda National Council for Science and Technology, and the University of California San Francisco Committee on Human Research approved the study protocol.

## Results

From October to December 2018, 1,482 (83%) adults presented to local clinics with recruitment cards for evaluation, including HIV testing. Of 1,482 assessed for eligibility, 957 (65%) did not meet inclusion criteria: The most common reasons for ineligibility were baseline HIV–positive status (34% [334/957]), reporting none of the HIV risk factors at screening (34% [322/957]), and reporting frequent testing in the prior 12 months (21% [204/957]; Fig 2).

Overall, 525 participants were randomized to financial incentives ($N = 173$), deposit contracts ($N = 172$), or no incentives ($N = 180$; Fig 2). One participant in the financial incentives group was determined to be ineligible post-randomization due to a false-negative baseline HIV test and was withdrawn. Participants' median age was 25 years (IQR: 22 to 30), 231 (44%) were women, and median weekly income was US$13.60 (IQR: US$8.16 to US$21.76). Baseline demographic characteristics, weekly income, and self-reported HIV risk factors of participants did not differ significantly across study groups, apart from a higher proportion of the deposit contract group (8%) having completed more than secondary school than the incentive (2%) and control (3%; $p = 0.03$) groups (Table 1).

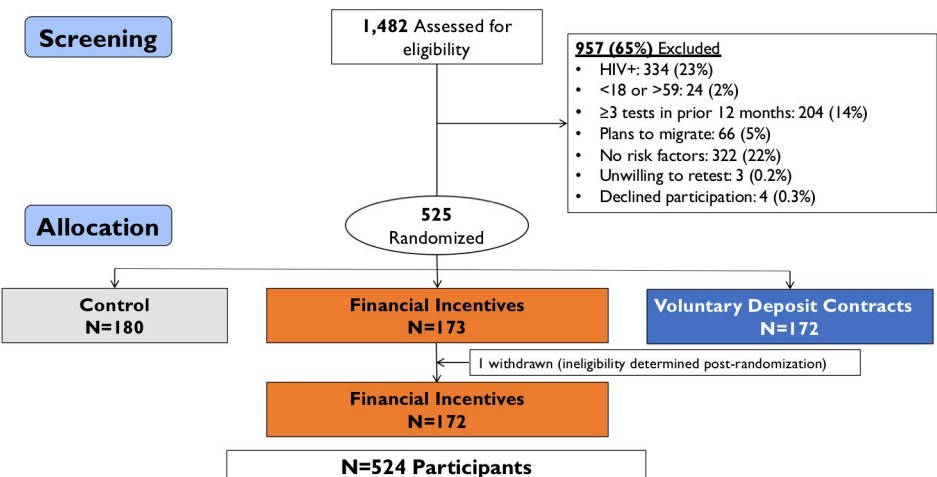

**Fig 2. Participant flowchart indicating screening, randomization, and allocation to study group in a randomized controlled trial of financial incentives and deposit contracts to promote HIV retesting in Uganda.**

**Table 1. Baseline characteristics of participants by study group in a randomized trial of financial incentives and deposit contracts to promote HIV retesting.**

| Baseline Data | Control N (%) | Incentive N (%) | Deposit N (%) |
|---|---|---|---|
| Enrolled | **180 (34)** | **172 (33)** | **172 (33)** |
| Recruitment Site | | | |
| Bars | 75 (42) | 79 (46) | 73 (42) |
| *Boda boda* stages[a] | 83 (46) | 79 (46) | 82 (48) |
| Other | 22 (12) | 14 (8) | 17 (10) |
| Median age (IQR) | 25 (21–31) | 25 (22–30) | 25 (22–29) |
| Sex | | | |
| Male | 100 (56) | 96 (56) | 97 (56) |
| Female | 80 (44) | 76 (44) | 75 (44) |
| Marital status | | | |
| Married/cohabitating | 80 (44) | 80 (46) | 77 (45) |
| Divorced/widowed | 38 (21) | 39 (23) | 38 (22) |
| Never married | 62 (34) | 53 (31) | 57 (33) |
| Highest school completed | | | |
| ≤Primary | 172 (96) | 164 (95) | 153 (89) |
| Secondary | 3 (2) | 5 (3) | 5 (3) |
| Tertiary | 5 (3) | 3 (2) | 14 (8) |
| Occupation | | | |
| Bar owner/worker | 70 (39) | 63 (37) | 61 (35) |
| Boda/motorcycle driver | 80 (44) | 77 (45) | 81 (47) |
| Other | 30 (17) | 32 (19) | 30 (17) |
| Median weekly income in Ugandan Shillings [US$[b]], (IQR) | 50,000 [US$13.76] (27,500–80,000) | 50,000 [US$13.76] (25,000–80,000) | 50,000 [US$13.76] (32,500–80,000) |
| Risk factors in prior 12 months, by self-report[c] >1 sexual partner HIV+ sexual partner STI diagnosis Transactional sex[d] | 172 (96%) 19 (11%) 55 (31%) 111 (62%) | 167 (97%) 19 (11%) 50 (29%) 109 (63%) | 168 (98%) 19 (11%) 55 (32%) 113 (66%) |

a "boda boda": local term for motorcycle taxi.

b 2018 US Dollars.

c Not mutually exclusive risk factors (participants could report >1 risk factor).

d Either paid or received money/gifts in exchange for sex.

Among deposit contract group participants, 24/172 (14%) made a baseline deposit. The median age of these participants was higher than those who did not make a deposit (28 versus 24.5 years; $p = 0.02$). Median weekly income was higher among those who made baseline deposits (US$15.14) than who did not (US$13.76; $p = 0.74$); but this difference was not statistically significant. Two (1%) participants in the deposit contract group made a deposit at 3 months.

In intent-to-treat analyses, HIV retesting at both 3 and 6 months was significantly higher in the incentive group (52% [89/172]) than either the deposit contract group (16% [28/172], OR 5.5, 95% CI: 3.3 to 9.1, $p < 0.001$]) or control group (18% [33/180], OR 4.8, 95% CI: 3.0 to 7.7, $p < 0.001$; Fig 3). There was no significant difference between HIV retesting in the deposit contract and control groups (OR 0.87 retesting in deposit group, 95%CI: 0.5 to 1.5, $p = 0.6$). Overall, a higher proportion of participants in all groups retested at 3 months (267/524 [51%]) than 6 months (165/524 [31%]). Nonetheless, at both times, HIV retesting was significantly higher in the financial incentive group than the deposit contract and control groups (Fig 3).

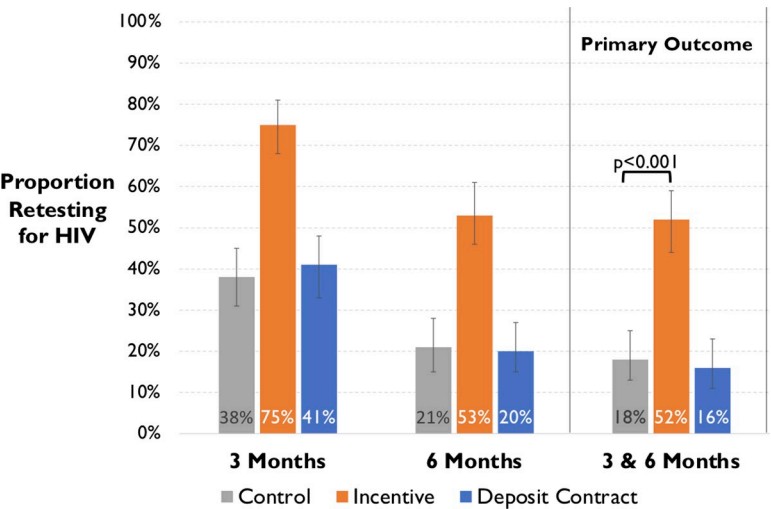

**Fig 3. The proportion of participants (with 95% confidence intervals) retesting for HIV at both 3- and 6-month post-randomization (primary outcome) and at the 3-month or 6-month time points, by study group in a randomized trial of financial incentives and deposit contracts vs control, to promote HIV retesting.**

Within the deposit contract group, those who made a baseline deposit were significantly more likely to retest at both 3 and 6 months than those who did not make a deposit (11/24 [46%] versus 17/148 [11%], *p* < 0.001. Fig 4). Of those who made a baseline deposit, 20/24 (83%) returned for their deposits and retested for HIV at 3 months. Of the 2 participants who made a deposit at 3 months, both returned and retested at 6 months. In instrumental variables analysis, however, there was no statistically significant effect of making a baseline deposit on HIV retesting at 3 months or 6 months (S1 Table).

Among 282 participants who retested for HIV during the trial, three (1%; 95%CI: 0.2% to 3%) seroconverted; all tested HIV–positive at the 3-month visit. One participant who

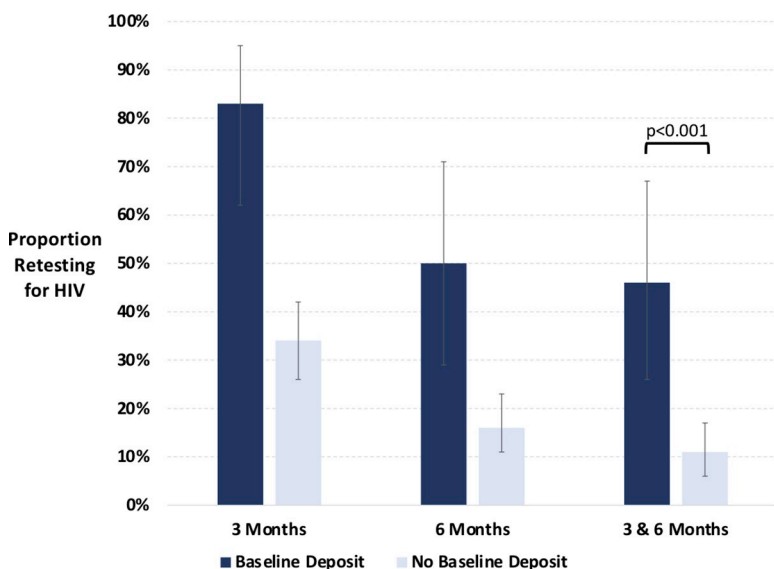

**Fig 4. The proportion of participants (with 95% confidence intervals) retesting for HIV at 3 months, 6 months, and both 3 and 6 months within the deposit contract group, stratified by those who did or did not make a baseline deposit.**

seroconverted was in the incentive group, and two were in the control group. All 3 participants were referred to HIV care and started ART on the day of testing HIV–positive. Trial activities for the final participants were completed on August 7, 2019.

## Discussion

In this randomized controlled trial of financial incentives and deposit contracts to promote routine retesting for HIV among adults at increased risk of HIV in Uganda, providing financial incentives led to a 2.9-fold (52% versus 18%) increase in retesting compared to the control of counseling on HIV retesting. In contrast, deposit contracts did not increase retesting rates. Given the need for routine HIV retesting among the many persons at increased risk of HIV who have previously tested negative, results from this study have several implications. Most notably, financial incentives should receive strong consideration as a strategic approach to increase retesting in priority populations. Cost-effectiveness modeling could help inform costs and impacts of such an approach in different settings. Additionally, although deposit contracts were feasible and associated with very high HIV retesting among participants who made deposits, their overall effectiveness was limited by the low number of individuals willing to make a deposit. Our findings add to the evidence on the effectiveness of financial incentives in promoting health behaviors and provide new evidence on other interventions informed by behavioral economics such as deposit contracts.

Routine retesting in persons at increased risk of HIV infection is critical to early HIV diagnosis. If followed by prompt ART initiation and viral suppression, early diagnosis reduces morbidity and mortality [1] and can eliminate onward HIV transmission [2]. Routine resting also offers opportunities to increase access to the latest HIV prevention tools [29]. We were able to rapidly enroll members of key populations using a simple community-based recruitment strategy informed by community leader input, as demonstrated by the high HIV positivity among persons screened. Furthermore, we observed a high proportion of new HIV infections among participants who retested (1%) within a 6-month period, emphasizing the need for retesting high-risk populations. Notably, despite the risk behaviors reported by adults screened, few (14%) reported routine retesting as recommended by Ugandan guidelines.

Few studies have rigorously evaluated strategies to promote HIV retesting among key populations in sub-Saharan Africa [10]. In a trial of direct provision of several HIV self-tests at one time compared to facility-based testing or facility-based HIV self-testing (HIVST) among FSW in Uganda, provision of self-tests increased retesting over 4 months [30]. In another study in family planning clinics in Uganda, integrating HIV testing resulted in a significantly higher proportion of clinic patients testing at least 3 times over 1 year compared to clinics that did not integrate testing [31]. Lastly, a trial in Kenya that randomized 18- to 29-year-olds at risk for acute HIV to either a standard appointment or an appointment with a text and phone call reminder found that reminders significantly increased retesting 2 to 4 weeks following an initial negative HIV test [32]. Our results demonstrate that financial incentives are also an effective strategy to increase retesting among persons at increased risk of HIV.

Several studies have demonstrated the effectiveness of financial incentives in promoting one-time HIV testing and other health-related behaviors [15,16,21,33]. Our study adds to the literature by showing how ongoing use of incentives can promote repeated behaviors that tend to decline over time. For example, several studies have found large declines in preexposure prophylaxis (PrEP) adherence and clinic engagement over time in sub-Saharan Africa [34–36]. Though our trial took place before widespread PrEP implementation in Uganda, effective strategies to promote retesting, if offered alongside the choice of PrEP and other emerging prevention strategies [3], could be used to engage those who may not consider themselves at risk

for retesting and prevention services. Of note, we observed a decline in testing at 6 months across all groups, suggesting additional interventions may be needed for sustained behavior change in our study population.

Voluntary deposit contracts are a promising approach for promoting behavior change because they directly address present bias in decision-making and leverage loss aversion. They may also be less expensive to implement since participants' put their own money at risk. Deposit contracts have been studied for behaviors such as weight loss and smoking cessation in high- and middle-income countries [20–23] but have not been implemented or evaluated, to our knowledge, in low-income settings where poverty may limit individuals' ability to make deposits. We attempted to overcome this barrier by offering deposit contracts during the same visits in which participants had received half the deposit amount as compensation for coming to the clinic, thus making it easier for participants to make a deposit while still leveraging loss aversion. In a prior pilot study, we observed that a much higher proportion of participants (>90%) were willing to make baseline deposits when the deposit amount was equal to or less than the incentive for baseline testing [24]. In this study, we increased the deposit amount in order to require a larger precommitment of one's own money and thus generate a greater sense of loss aversion. However, perhaps as a consequence, we observed relatively low baseline deposit contract uptake (14%). We suspect that had our deposit amount been lower, we would have observed higher baseline deposit contract uptake, but possibly also lower testing uptake among those making deposits. Our findings of low deposit contract uptake are similar to trials of deposit contracts for smoking cessation that have observed uptake ranging from 11% to 13.7% [20,23]. Importantly, although deposit contracts did not result in increased HIV retesting overall, participants who made deposits retested at extremely high levels, suggesting that for some, the decision to precommit to future testing may have been motivating. Alternatively, those with the greatest motivation to retest may have been more likely to make deposits. Future research could consider comparing differing deposit contract amounts and interest earned to increase participation while maintaining the potential of leveraging precommitment and loss aversion for behavior change.

Our study has limitations. First, we measured HIV retesting at clinics where baseline enrollment occurred: If participants opted to retest elsewhere, we may have undermeasured retesting. However, given the low rate of routine HIV retesting reported at baseline, we suspect this was unlikely to have impacted our results. Second, we only offered facility-based testing. Whether such strategies could increase retesting at out-of-facility venues is not clear. In the context of the COVID-19 pandemic, considering incentives for retesting, in combination with access to HIVST or non-facility-based testing venues, may allow programs to avoid losing ground on HIV retesting among at-risk persons and merits further evaluation. Lastly, we did not evaluate retesting beyond the 6-month trial period or post-trial retesting behavior, and whether incentives may have resulted in any habit formation with durable impact on retesting behavior, or conversely undermined intrinsic motivation to retest, after incentives were no longer available is unknown. Despite these limitations, our study provides rigorous evidence that financial incentives can significantly increase HIV retesting among high-risk adults.

In conclusion, this study tests novel interventions to promote HIV retesting and finds that financial incentives lead to large and significant increases in retesting. Deposit contracts, which leverage behavioral economics principles more strongly and are less costly than financial incentives, do not increase retesting rates overall even though they result in high retesting among those who precommit to retesting by making a deposit. As efforts to end HIV by 2030 increasingly rest on early HIV diagnosis among high-risk populations, strategic use of incentives to promote retesting should receive greater consideration by HIV programs.

## Supporting information

**S1 Text. CONSORT checklist for randomized controlled trials.**
(PDF)

**S2 Text. Study protocol.**
(PDF)

**S1 Table. Instrumental variable regression results to estimate causal effect of making a deposit on HIV retesting.**
(DOCX)

## Acknowledgments

We gratefully acknowledge our research staff, community advisory board members, and especially the communities and participants involved in this study.

## Author Contributions

**Conceptualization:** Gabriel Chamie, Harsha Thirumurthy.

**Data curation:** Kara Marson.

**Formal analysis:** Harsha Thirumurthy.

**Funding acquisition:** Gabriel Chamie, Harsha Thirumurthy.

**Investigation:** Gabriel Chamie, Dalsone Kwarisiima, Alex Ndyabakira, Kara Marson, Carol S. Camlin, Diane V. Havlir, Moses R. Kamya, Harsha Thirumurthy.

**Methodology:** Gabriel Chamie, Dalsone Kwarisiima, Alex Ndyabakira, Harsha Thirumurthy.

**Project administration:** Dalsone Kwarisiima, Alex Ndyabakira, Kara Marson.

**Supervision:** Gabriel Chamie, Dalsone Kwarisiima, Alex Ndyabakira, Kara Marson, Harsha Thirumurthy.

**Writing – original draft:** Gabriel Chamie.

**Writing – review & editing:** Dalsone Kwarisiima, Alex Ndyabakira, Kara Marson, Carol S. Camlin, Diane V. Havlir, Moses R. Kamya, Harsha Thirumurthy.

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
