## [Editor Report · Decision Letter 0]

7 Jan 2021

Dear Dr Chamie, 

Thank you for submitting your manuscript entitled "A Randomized Trial of Financial Incentives and Deposit Contracts to Promote HIV Retesting" for consideration by PLOS Medicine.

Your manuscript has now been evaluated by the PLOS Medicine editorial staff and I am writing to let you know that we would like to send your submission out for external assessment. 

Once your full submission is complete, your paper will undergo a series of checks in preparation for external assessment.

Kind regards,

Richard Turner, PhD

rturner@plos.org

---

## [Decision Letter · Decision Letter 1]

2 Mar 2021

Dear Dr. Chamie,

Thank you very much for submitting your manuscript "A Randomized Trial of Financial Incentives and Deposit Contracts to Promote HIV Retesting" (PMEDICINE-D-21-00078R1) for consideration at PLOS Medicine. 

Your paper was evaluated by an academic editor with relevant expertise and sent to independent reviewers, including a statistical reviewer. The reviews are appended at the bottom of this email and any accompanying reviewer attachments can be seen via the link below:

[LINK]

In light of these reviews, we will not be able to accept the manuscript for publication in the journal in its current form, but we would like to invite you to submit a revised version that addresses the reviewers' and editors' comments fully. You will appreciate that we cannot make a decision about publication until we have seen the revised manuscript and your response, and we expect to seek re-review by one or more of the reviewers. 

We hope to receive your revised manuscript by Mar 23 2021 11:59PM. Please email us (plosmedicine@plos.org) if you have any questions or concerns.

Please let me know if you have any questions, and we look forward to receiving your revised manuscript. 

Sincerely,

Richard Turner, PhD

rturner@plos.org

Please adapt your title so that the study descriptor ("a randomized trial" falls after a colon); and include the country name.

In the abstract and results section, please quote effect sizes and 95% CI for the primary endpoint findings.

Please quote study dates in your abstract.

Please add a few final sentence to the "Methods and findings" subsection of your abstract, beginning "Study limitations include ..." or similar and quoting 2-3 of the study's main limitations. 

After the abstract, we will need to ask you to add a new and accessible "Author summary" section in non-identical prose. You may find it helpful to consult one or two recent research papers published in PLOS Medicine to get a sense of the preferred style. 

Please refer to figure 2 as the "Participant flowchart" or similar, rather than "CONSORT diagram". 

We believe that CONSORT discourages statistical tests at baseline in randomized trials, and ask that you remove these from table 1. 

At line 261 and any other instances in the paper, please avoid "nearly three-fold", instead quoting actual numbers. 

Throughout the text, please style reference call-outs as follows: "... HIV transmission [1,2]. Similarly ... " (noting the absence of spaces within the square brackets). 

Please remove the information on study funding from the end of the main text. In the event of publication, this information will appear in the article metadata, via entries in the submission form. 

In the reference list, please ensure that all references have full access details, e.g., reference 15. 

Please ensure that journal names are abbreviated consistently. 

Please add a completed CONSORT checklist, labelled "S1_CONSORT_Checklist" or similar and referred to as such in the Methods section. In the checklist, please refer to individual items by section (e.g., "Methods") and paragraph number rather than by page or line numbers, as the latter generally change in the event of publication. 

Please include the study protocol as a supplementary document, referred to in your Methods section, unless this is published. 

Comments from the reviewers:

*** Reviewer #1: 

[See attachment]

Michael Dewey

*** Reviewer #2: 

This is an interesting, well conducted and well reported study. 

My main comment is about the fact that the take-up for the deposit contract appeared to be quite low. I was not necessarily surprised by this finding given the low average income in the study setting and therefore the predictably low ability to save. I am surprised that this did not appear as a major constraint for the feasibility of study. Specifically, I would have liked the authors to compare their finding in their pilot study (Chamie G, Ndyabakira A, Marson KG, Emperador DM, Kamya MR, Havlir DV, et al. A pilot randomized trial of incentive strategies to promote HIV retesting in rural Uganda. PLoS ONE. 2020;15(5):e0233600.) with their findings in the present study. Did they manage to get higher uptake of the deposit contracts in the pilot study? Apparently yes, since in the pilot study 93% made deposits, but only 14% in the current study. Why and what might explain the differences in uptake between the pilot study and current study? This seems to be a key point to discuss.

Given the low uptake for the deposit contracts in the current study, very little can be concluded about the comparison between cash incentives and deposit contracts, which, I suppose, was the main objective of the study. The remaining result is about the effectiveness of the cash incentives, but that effectiveness has already been established (see Lee R, Cui RR, Muessig KE, Thirumurthy H, Tucker JD. Incentivizing HIV/STI Testing: A Systematic Review of the Literature. AIDS Behav. 2013.)

Minor comment:

I am surprised not to have found one of the first study testing incentives for HIV testing in the reference list:

Thornton RL. The Demand for, and Impact of, Learning HIV Status. Am Econ Rev. 2008;98(5):1829-63.

*** Reviewer #3: 

A Randomized Trial of Financial Incentives and Deposit Contracts to Promote HIV Retesting

Manuscript Number: PMEDICINE-D-21-00078R1

This manuscript reports the results of a three-group randomized, controlled trial to determine the effectiveness of 1) financial incentives and 2) deposit contracts vs. 3) control in achieving HIV re-testing at both 3 and 6 months post-randomization among people at high risk for contracting HIV in southwestern Uganda. Eligible individuals were those who presented for an evaluation and who were 18-59 years of age, tested negative for HIV, and reported at least one of the following risk factors in the prior 12 months: 1) >1 partner; 2) a known HIV-infected partner; 3) a history of a sexually transmitted infection; or 4) paid or received money or gifts in exchange for sex. Self-report (at screening/baseline) and HIV testing data (screening/baseline and follow-up) were collected over a ~6-month observation period. 

Strengths of this study are its large sample size, three-arm randomized and controlled design, rapid recruitment period, and high-risk target population. The paper is well organized, clearly written and will add to the body of literature on the effectiveness of financial incentives and deposit contracts. This reviewer found no major issues and only a handful of minor issues (described below) that authors may consider addressing.

Major Issues: None

Minor Issues:

1) Lines 134-135: Consider including a brief description of measures assessed via the baseline questionnaire. If not feasible due to space limitations, consider referring readers to Table 1 for a list of the measures. Also, consider clarifying whether any follow-up questionnaire was administered at 3- and 6-month follow-ups.

2) Line 147: Figure 1 refers to "cash transfer", yet this term is not addressed within the text. Does this payment correspond to the "one-time reimbursement" of $2.70 paid to individuals who completed the screening/baseline evaluation (as mentioned in line 118)? 

 a) If yes, then I recommend using parallel terminology and $ amount in the text and figure to add clarity. 

 b) If no, then I recommend the term, "cash transfer", be briefly defined or described within the text prior to being used in the figure.

3) Line 188: The recruitment period is clearly describing. For clarity and context, consider clearly stating the full study observation period and/or "stop date" (last date on which data were collected for last randomized participant). 

4) Lines 265-266: The phrase, "Further cost-effectiveness modeling…" sounds like some cost-effectiveness modeling may already have been performed and additional modeling is recommended. Is this the case?

 a) If yes, please briefly present results of any cost-effectiveness modeling already performed. 

 b) If no, please eliminate "Further" and start the sentence with "Cost-effectiveness modeling…" to avoid confusion.

5) Line 332: Consider addressing the short observation period (only 6 months) as a study limitation, particularly because lines 66-67 indicate that Uganda Ministry of Health recommends HIV retesting every 3 months for "key populations" which (presumably) includes the high-risk individuals recruited for this study.

***

[LINK]

---

## [Decision Letter · Decision Letter 2]

10 Apr 2021

Dear Dr. Chamie,

Thank you very much for re-submitting your manuscript "Financial Incentives and Deposit Contracts to Promote HIV Retesting in Uganda: a randomized trial" (PMEDICINE-D-21-00078R2) for consideration at PLOS Medicine.

I have discussed the paper with our academic editor and it was also seen again by three reviewers. I am pleased to tell you that, provided the remaining editorial and production issues are dealt with, we expect to be able to accept the paper for publication in the journal.

[LINK]

Please let me know if you have any questions, and we look forward to receiving the revised manuscript shortly.   

Sincerely,

Richard Turner, PhD

rturner@plos.org

Requests from Editors:

Please finalize the arrangements for data deposition and release. 

Please add a sentence, say, at line 51 to quote the number, and distribution by study arm, of seroconversions. 

Throughout the text, please move reference call-outs before punctuation (e.g., " ... HIV transmission [1,2].").

In the reference list, please abbreviate journal names consistently (e.g., "PLoS Med.").

Comments from Reviewers:

***Reviewer #1: 

The authors have addressed my points and have clearer up the point about the randomisation.

Michael Dewey

*** Reviewer #2: 

Thank you for your responses to my comments.

I remain convinced of the relevance of my second comment:

"Given the low uptake for the deposit contracts in the current study, very little can be concluded

about the comparison between cash incentives and deposit contracts, which, I suppose, was the

main objective of the study. The remaining result is about the effectiveness of the cash

incentives, but that effectiveness has already been established (see Lee R, Cui RR, Muessig KE,

Thirumurthy H, Tucker JD. Incentivizing HIV/STI Testing: A Systematic Review of the Literature.

AIDS Behav. 2013.) "

Your answer is technically correct but does not convince me that the results from this randomized control trial are sufficiently novel and important to be published in PLoS Medicine.

*** Reviewer #3: 

The "minor issues" that I raised were satisfactorily addressed in this revision.

***

[LINK]

---

## [Editor Report · Decision Letter 3]

15 Apr 2021

Dear Dr Chamie, 

On behalf of my colleagues and the Academic Editor, Dr Barnabas, I am pleased to inform you that we have agreed to publish your manuscript "Financial Incentives and Deposit Contracts to Promote HIV Retesting in Uganda: a randomized trial" (PMEDICINE-D-21-00078R3) in PLOS Medicine.

PRESS

Sincerely, 

Richard Turner, PhD 

rturner@plos.org